# Liraglutide Attenuates FFA-Induced Retinal Pigment Epithelium Dysfunction via AMPK Activation and Lipid Homeostasis Regulation in ARPE-19 Cells

**DOI:** 10.3390/ijms26083704

**Published:** 2025-04-14

**Authors:** Sing-Hua Tsou, Kai-Shin Luo, Chien-Ning Huang, Edy Kornelius, I-Ting Cheng, Hui-Chih Hung, Yu-Chien Hung, Chih-Li Lin, Min-Yen Hsu

**Affiliations:** 1Department of Medical Research, Chung Shan Medical University Hospital, Taichung 402306, Taiwan; zinminid@gmail.com; 2Institute of Medicine, Chung Shan Medical University, Taichung 402306, Taiwan; seekcoeur@gmail.com (K.-S.L.); cshy049@csmu.edu.tw (C.-N.H.); korn3lius82@gmail.com (E.K.); hchung@dragon.nchu.edu.tw (H.-C.H.); 3School of Medicine, Chung Shan Medical University, Taichung 402306, Taiwan; 4Department of Ophthalmology, Chung Shan Medical University Hospital, Taichung 402306, Taiwan; b92401086@gmail.com; 5Department of Internal Medicine, Division of Endocrinology and Metabolism, Chung Shan Medical University Hospital, Taichung 402306, Taiwan; 6Department of Medical Laboratory and Biotechnology, Chung Shan Medical University, Taichung 402306, Taiwan; tina2397828@gmail.com; 7Department of Life Sciences, Institute of Genomics and Bioinformatics, National Chung Hsing University, Taichung 402202, Taiwan

**Keywords:** retinal pigment epithelium (RPE), lipid droplet (LD), free fatty acid (FFA), epithelial-mesenchymal transition (EMT), liraglutide, AMP-activated protein kinase (AMPK)

## Abstract

Age-related macular degeneration (AMD) is a leading cause of irreversible vision loss in the elderly, and it is characterized by oxidative stress, lipid dysregulation, and dysfunction of the retinal pigment epithelium (RPE). A hallmark of AMD is the presence of drusen, extracellular deposits rich in lipids, proteins, and cellular debris, which are secreted by the RPE. These deposits impair RPE function, promote chronic inflammation, and accelerate disease progression. Despite advancements in understanding AMD pathogenesis, therapeutic strategies targeting lipid dysregulation and oxidative damage in RPE cells remain limited. This study evaluated the effects of liraglutide, a glucagon-like peptide-1 receptor agonist (GLP-1RA), on free fatty acid (FFA)-induced damage in ARPE-19 cells, a widely used in vitro model of RPE dysfunction. FFA treatment induced lipid droplet accumulation, oxidative stress, and epithelial–mesenchymal transition (EMT), which are processes implicated in AMD progression. Liraglutide significantly reduced lipid droplet accumulation, mitigated oxidative stress, and suppressed EMT, as demonstrated by high-content imaging, immunocytochemistry, and molecular assays. Mechanistic analyses revealed that liraglutide activates AMP-activated protein kinase (AMPK), enhancing lipophagy and restoring lipid homeostasis. Furthermore, liraglutide influenced exosome secretion, altering paracrine signaling and reducing EMT markers in neighboring cells. These findings underscore liraglutide’s potential to address critical mechanisms underlying AMD pathogenesis, including lipid dysregulation, oxidative stress, and EMT. This study provides foundational evidence supporting the development of GLP-1 receptor agonists as targeted therapies for AMD.

## 1. Introduction

Age-related macular degeneration (AMD) is a leading cause of vision loss among older adults and represents a significant global health concern due to its impact on quality of life and its increasing prevalence in aging populations. Central to AMD progression is the dysfunction of retinal pigment epithelium (RPE) cells, which play a vital role in supporting retinal health. The RPE forms a monolayer beneath the retina and is responsible for transporting nutrients, clearing photoreceptor waste, and managing oxidative stress [1]. When these cells become dysfunctional, retinal integrity deteriorates, emphasizing the critical importance of RPE health in maintaining vision. RPE dysfunction in AMD arises from oxidative stress, chronic inflammation, and metabolic imbalance [2]. Oxidative damage impairs waste processing, leading to the buildup of harmful byproducts such as lipofuscin [3]. This disruption contributes to the formation of drusen—extracellular deposits rich in lipids and proteins—which are early indicators of AMD [4]. Over time, these changes compromise the balance between the retina and the underlying choroid, progressing to advanced forms of AMD, including geographic atrophy or choroidal neovascularization, both of which severely impair vision [5]. Current therapeutic strategies focus on reducing oxidative damage, enhancing cellular waste clearance mechanisms such as autophagy, and mitigating lipid accumulation in RPE cells to slow disease progression and preserve vision [6].

Lipid metabolism disturbances are particularly detrimental to RPE function and contribute significantly to AMD progression [7]. Elevated levels of free fatty acids (FFAs) in RPE cells lead to the accumulation of lipid droplets (LDs), inducing oxidative and endoplasmic reticulum stress [8]. FFAs are also implicated in the formation of drusen due to their role in the lipid-rich composition of these deposits [9]. Additionally, FFAs drive epithelial–mesenchymal transition (EMT), causing RPE cells to lose their structural integrity and epithelial properties while adopting migratory behaviors [10]. This disrupts cell-to-cell adhesion, compromises the integrity of the RPE barrier—also known as the blood–retina barrier (BRB)—and promotes inflammation and fibrosis in the retina, further impairing retinal health [11]. FFAs also impair lipophagy, a critical autophagic process that clears excess lipids, leading to further metabolic imbalance [12,13]. Collectively, these processes exacerbate RPE dysfunction, accelerating AMD progression and underscoring the need for therapies that address lipid dysregulation [14].

Glucagon-like peptide-1 receptor agonists (GLP-1 RAs), initially developed for type 2 diabetes, show promise in addressing metabolic and degenerative diseases. GLP-1 receptors (GLP-1R) are expressed in retinal pigment epithelial (RPE) cells, as demonstrated in both in vitro and in vivo studies, including immunohistochemical analyses in diabetic Goto-Kakizaki rats [15,16]. GLP-1 RAs have been shown to regulate glucose and lipid metabolism, reduce oxidative stress, and promote autophagy [17]. By activating AMP-activated protein kinase (AMPK), GLP-1 RAs can reduce lipid accumulation and restore cellular homeostasis [18]. Excessive FFA levels in RPE cells not only induce oxidative stress but also promote cellular senescence, which accelerates lipid dysregulation and further damages RPE cells [19,20]. These findings suggest that GLP-1 signaling may offer a therapeutic approach to mitigate the metabolic imbalances driving AMD. In this study, we investigate the potential of liraglutide, a GLP-1 RA, to protect RPE cells from FFA-induced damage. Specifically, we evaluate its ability to reduce lipid droplet accumulation, mitigate EMT, enhance lipophagy, alleviate oxidative stress, and regulate exosome secretion. By addressing these interconnected pathological mechanisms, this research aims to provide new insights into the therapeutic potential of GLP-1 RAs for AMD and to highlight the role of GLP-1 signaling in maintaining RPE health.

## 2. Results

### 2.1. Liraglutide Reduces FFA-Induced LD Accumulation in ARPE-19 Cells

To identify optimal treatment conditions, an MTT assay evaluated the effects of free fatty acids (oleic acid: palmitic acid = 2:1) and liraglutide on ARPE-19 cells. As shown in Figure 1A, 0.1 μM liraglutide alone did not affect cell viability after 24 h. FFA concentrations of 100 μM and 250 μM had no significant impact on viability, whereas 500 μM markedly reduced survival. Consequently, 250 μM FFA was selected for further experiments. When combined with 250 μM FFA, liraglutide at 0.05 μM and 0.1 μM showed no adverse effects on viability, while 0.5 μM liraglutide significantly reduced survival. Thus, 0.1 μM liraglutide was chosen as the optimal concentration. Bright-field microscopy revealed no observable changes in cell morphology or survival with either 250 μM FFA or 0.1 μM liraglutide alone. Combined treatment also showed no morphological alterations, further supporting the selected concentrations (Figure 1B). LipidTox staining demonstrated that FFA significantly increased intracellular lipid droplet accumulation, while co-treatment with 0.1 μM liraglutide effectively reduced this accumulation (Figure 1C). The quantitative analysis of LDs was performed using high-content analysis (HCA), a computer-assisted imaging method for the automated counting and statistical assessment of lipid droplets. This approach accurately distinguished LDs of different sizes, ensuring the objective and reliable analysis of cellular responses. As shown in Figure 1D, FFA treatment significantly increased the number of LDs larger than 1 μm in diameter. Co-treatment with liraglutide reduced both the total number of LDs and the number of larger LDs while increasing the number of smaller LDs (<1 μm). These results suggest that liraglutide enhances the degradation of larger lipid droplets, potentially improving lipid metabolism in ARPE-19 cells. In summary, liraglutide effectively reduces FFA-induced lipid droplet accumulation in ARPE-19 cells, likely by promoting lipid breakdown. These findings highlight its potential for addressing lipid dysregulation in RPE cells and its relevance to AMD pathology.

### 2.2. Liraglutide Suppresses FFA-Induced EMT and Restores Tight Junction Integrity in ARPE-19 Cells

Figure 2 illustrates the protective effects of liraglutide against FFA-induced tight junction disruption and EMT in ARPE-19 cells. The immunocytochemistry staining of ZO-1, a tight junction marker (Figure 2A), showed a significant reduction in membrane-associated ZO-1 expression following 24 h of FFA treatment. This reduction was particularly evident in regions with elevated LD accumulation, as confirmed by LipidTox staining. Co-treatment with liraglutide restored ZO-1 expression to control levels, indicating its ability to maintain tight junction integrity. Western blot analysis further confirmed these findings (Figure 2B). FFA treatment significantly decreased the epithelial marker E-cadherin and increased the mesenchymal marker vimentin and phosphorylated Smad2/3 (pSer-Smad2/3), key regulators of EMT. Liraglutide reversed these effects by restoring E-cadherin levels and reducing vimentin and pSer-Smad2/3 expression. These trends were corroborated by the ICC staining of E-cadherin and vimentin (Figure 2C), which showed the following consistent results: FFA reduced E-cadherin and increased vimentin, both of which were reversed with liraglutide co-treatment. The quantitative PCR analysis of EMT-related transcription factors (Figure 2D) demonstrated that FFA treatment significantly upregulated Snail, Twist1, Twist2, and Slug, further confirming its role in EMT induction. Liraglutide co-treatment effectively suppressed the expression of these transcription factors, indicating its ability to counteract EMT at the transcriptional level. Together, these results demonstrate that liraglutide reduces FFA-induced EMT and preserves tight junction integrity in ARPE-19 cells, underscoring its potential therapeutic role in mitigating AMD-associated epithelial dysfunction.

### 2.3. Liraglutide Enhances Lipophagy to Promote LD Degradation in ARPE-19 Cells

Figure 3 highlights the effects of liraglutide in counteracting the FFA-induced suppression of autophagy and promoting lipophagy, facilitating LD degradation in ARPE-19 cells. Acridine orange (AO) staining (Figure 3A) revealed a significant reduction in acidic vesicular organelles (AVOs) following FFA treatment, indicating inhibited autophagy. Liraglutide co-treatment restored AVO expression, suggesting enhanced autophagic activity. This finding was supported by HCA (Figure 3B), which quantified AVO numbers. FFA reduced AVOs to approximately 10% of control levels, while liraglutide increased AVO numbers to 50% of control levels. Western blot analysis (Figure 3C) further confirmed these observations. FFA significantly decreased phosphorylated AMPK levels, reduced LC3-II expression, and increased p62 accumulation, consistent with suppressed autophagy. Liraglutide co-treatment restored phosphorylated AMPK levels, increased LC3-II expression, and reduced p62 levels, indicating the activation of AMPK-mediated autophagy. In addition, we evaluated the activity of mTOR, a key downstream effector negatively regulated by AMPK. FFA treatment increased phosphorylated mTOR levels, indicating suppressed autophagy and metabolic dysfunction. Liraglutide significantly reduced mTOR phosphorylation, supporting the notion that liraglutide relieves FFA-induced autophagy suppression through the AMPK/mTOR signaling axis. These results suggest that modulation of the mTOR pathway may contribute to the restoration of lipid homeostasis and to improved autophagic activity in RPE cells. To evaluate the role of lipophagy, perilipin 2 (PLIN2), a critical regulator of LD-specific autophagy, was analyzed using immunocytochemistry (Figure 3D). FFA treatment markedly reduced PLIN2 expression, whereas liraglutide restored it. The co-localization of PLIN2 with LDs was observed, indicating that liraglutide facilitates lipophagy and enhances LD degradation. Additionally, qPCR analysis (Figure 3E) demonstrated that FFA significantly downregulated lipophagy-associated genes, including autophagy related 5 (ATG5), as well as lysosomal fatty acid degradation genes such as lysosomal acid lipase (LIPA), ras-related protein 7A (RAB7A), and lysosomal-associated membrane protein 2 (LAMP2). Liraglutide co-treatment restored the expression of these genes, further supporting its role in promoting LD degradation pathways. These results implied that liraglutide alleviates the FFA-induced suppression of autophagy and lipophagy by activating AMPK signaling and upregulating genes that are essential for LD degradation. These findings underscore its therapeutic potential in mitigating lipid dysregulation associated with AMD.

### 2.4. AMPK Activation Is Essential for Liraglutide-Mediated Reduction of ROS and Lipid Accumulation in ARPE-19 Cells

Figure 4 illustrates that AMPK activation is critical for liraglutide’s ability to mitigate FFA-induced oxidative stress, lipid accumulation, and EMT in ARPE-19 cells. DCFH-DA staining (Figure 4A) revealed a substantial increase in intracellular ROS following FFA treatment, consistent with oxidative stress observed during AMD progression. Liraglutide markedly reduced ROS levels, but this effect was fully reversed by co-treatment with compound C (CC), an AMPK-specific inhibitor, underscoring the dependence of liraglutide’s ROS-lowering effects on AMPK activation. Quantitative fluorescence analysis (Figure 4B) confirmed these findings, showing that FFA increased ROS levels nearly seven-fold compared to controls, while liraglutide reduced this increase by approximately two-thirds. However, co-treatment with CC eliminated this reduction, further validating the requirement for AMPK activation. Western blot analysis (Figure 4C) demonstrated that liraglutide’s restoration of autophagy and modulation of EMT markers is AMPK-dependent. FFA treatment reduced phosphorylated AMPK and LC3-II expression while increasing p62 levels, which is indicative of suppressed autophagy. Concurrently, FFA decreased E-cadherin and increased vimentin, which is consistent with EMT induction. Liraglutide reversed these changes, restoring phosphorylated AMPK and LC3-II levels, reducing p62, and normalizing E-cadherin and vimentin expression. However, co-treatment with CC diminished liraglutide’s ability to modulate these markers, confirming the necessity of AMPK activation. Nile red staining (Figure 4D) further supported these findings, showing that liraglutide reduced intracellular lipid droplet accumulation induced by FFA, an effect that was completely negated by CC co-treatment. Collectively, these results demonstrate that liraglutide alleviates oxidative stress, lipid accumulation, and EMT in ARPE-19 cells through AMPK activation. The reversal of these protective effects by AMPK inhibition underscores its critical role in mediating liraglutide’s therapeutic potential for AMD-related cellular dysfunction.

### 2.5. Liraglutide Modulates Exosome Release and Size Distribution in ARPE-19 Cells Under FFA Stress

Figure 5 highlights the effects of liraglutide on exosome secretion, size distribution, and their role in EMT regulation under FFA-induced stress in ARPE-19 cells. Nanoparticle tracking analysis (NTA) (Figure 5A) revealed that FFA significantly increased extracellular vesicle (EV) secretion, with EV numbers approximately 10-fold higher than the control group (Figure 5B). Liraglutide partially reduced this increase, though EV levels remained around eight times higher than in controls. Co-treatment with GW4869, an inhibitor of exosome biogenesis and release, drastically reduced EV secretion, confirming that most of these EVs were exosomes secreted by ARPE-19 cells. Further analysis of particle size distribution (Figure 5C) showed that FFA increased the average diameter of exosomes. While liraglutide did not affect the average diameter, it significantly reduced the median diameter (D50) and the 90th percentile (D90), as well as the mode diameter of EVs. This indicates that liraglutide shifts the particle size distribution toward smaller exosomes, while the 10th percentile (D10) remained unchanged. To evaluate the functional impact of exosomes, EVs isolated from conditioned media were applied to FFA-treated ARPE-19 cells. To evaluate the functional impact of exosomes, EVs isolated from conditioned media under different treatment conditions were adjusted to a concentration of 5 × 10⁷ particles/mL and were co-cultured with FFA-treated ARPE-19 cells for 24 h. The immunocytochemistry analysis of epithelial (E-cadherin) and mesenchymal (vimentin) markers (Figure 5D) demonstrated that exosomes derived from the control and FFA + liraglutide groups enhanced E-cadherin expression and suppressed vimentin expression, indicating anti-EMT effects. Exosomes from the FFA group also reduced EMT, but their effect was weaker compared to those from the control and FFA + liraglutide groups. These findings highlight the ability of exosomes to influence EMT and suggest that liraglutide enhances their regulatory potential under FFA stress. In summary, liraglutide modulates exosome secretion and size distribution under FFA-induced stress and enhances the ability of exosomes to regulate EMT, demonstrating its potential to mitigate AMD-related RPE dysfunction.

## 3. Discussion

Drusen, hallmark features of AMD, play a central role in disease progression by impairing RPE function and promoting chronic inflammation. These extracellular deposits are composed of proteins, cellular debris, and lipids, with the latter contributing significantly to their formation [7]. The accumulation of neutral lipids within RPE cells leads to intracellular LD formation, which, when excessive, disrupts cellular homeostasis, induces oxidative stress, and interferes with essential processes such as lipid metabolism and EMT [21]. Consistent with these observations, our study found that FFA exposure significantly increased LD accumulation in ARPE-19 cells, mimicking the lipid dysregulation characteristic of AMD [22]. Importantly, treatment with liraglutide reduced LD accumulation, suggesting its potential to alleviate lipid overload and mitigate drusen formation. GLP-1 receptor activation has been shown to play a critical role in regulating lipid metabolism [23]. Previous studies have demonstrated that GLP-1 signaling promotes fatty acid oxidation, reduces lipid synthesis, and enhances lipid clearance across various cell types. Our findings align with this evidence, as liraglutide, a GLP-1 receptor agonist, reduced LD accumulation in FFA-treated RPE cells. This indicates that GLP-1 signaling may modulate lipid metabolism in RPE cells under stress conditions, restoring cellular homeostasis. Excessive LD accumulation is associated with oxidative stress, which disrupts lipid degradation pathways, perpetuates lipid buildup, and promotes inflammatory responses [24]. Furthermore, oxidative stress drives EMT, weakening RPE barrier integrity and contributing to AMD progression [25]. Our findings suggest that liraglutide may counteract these harmful processes by reducing LD accumulation and oxidative stress, potentially preserving RPE function and delaying AMD progression.

AMPK is a key regulator of cellular energy balance and a central pathway through which GLP-1 receptor agonists exert their effects [26]. AMPK activation has been shown to reduce lipid accumulation by suppressing lipid synthesis and enhancing lipid breakdown [27,28]. In this study, the inhibition of AMPK activity significantly diminished liraglutide’s protective effects, underscoring AMPK’s essential role in mediating the drug’s actions. Additionally, AMPK promoted lipophagy, a specialized autophagic process that degrades lipid droplets [29]. This mechanism may account for liraglutide’s ability to reduce lipid droplet accumulation in FFA-stressed RPE cells. Excessive lipid droplet accumulation has been linked to increased oxidative stress, which disrupts critical cellular functions, including lipid catabolism and EMT [30]. These disturbances exacerbate RPE dysfunction and contribute to the pathogenesis of AMD [31]. By restoring lipid homeostasis through AMPK activation, liraglutide may mitigate these deleterious effects. Although the specific physiological role of GLP-1 signaling in RPE cells remains unclear [32], our evidence indicates that activating this pathway significantly reduces lipid accumulation. These findings align with previous studies that highlight the potential of GLP-1 receptor agonists in regulating lipid metabolism [33], offering a promising therapeutic strategy for addressing lipid dysregulation in AMD. However, the safety of GLP-1 receptor agonists in retinal diseases remains controversial. A clinical trial reported an increased risk of diabetic retinopathy complications with semaglutide use [34], which is potentially linked to rapid glycemic improvement. Although this adverse effect may not be a class-wide phenomenon, these findings highlight the need for further investigation into the retinal safety profile of GLP-1-based therapies. Further research into the GLP-1/AMPK axis in RPE cells is warranted to provide deeper insights into its protective mechanisms and therapeutic applications.

The RPE plays a critical role in maintaining retinal homeostasis by supporting photoreceptors and facilitating intercellular communication, primarily through the secretion of cytokines and exosomes [35]. Exosomes, small extracellular vesicles, have gained increasing recognition for their ability to transfer bioactive molecules that regulate physiological and pathological processes in neighboring cells [36]. Under the oxidative stress and inflammatory condition characteristic of AMD, RPE exosome secretion is altered, potentially contributing to AMD progression by promoting retinal inflammation, angiogenesis, and cellular dysfunction [37]. Our findings indicate that liraglutide modulates RPE exosome secretion under FFA-induced stress. Specifically, exosomes from liraglutide-treated ARPE-19 cells suppressed EMT markers in neighboring FFA-treated cells, suggesting that liraglutide reduces EMT progression through exosome-mediated mechanisms. These results implicate GLP-1 receptor activation in altering the secretion dynamics or cargo composition of RPE-derived exosomes, which may mitigate intercellular signaling pathways that drive EMT and lipid dysregulation. In addition, although we did not analyze the molecular composition of exosomes in this study, our recent clinical findings suggest that exosome size may be closely related to the pathological state of retinal diseases [38]. Notably, larger exosomes were predominantly found in inflammatory diseases such as pars planitis, whereas smaller exosomes were more frequently associated with ischemic or neovascular conditions, including central retinal vein occlusion and vitreous hemorrhage. These observations indicate that exosome size heterogeneity may reflect different biological functions or disease mechanisms, which could be linked to variations in molecular cargo. While this study highlights a novel mechanism by which liraglutide may mitigate AMD-related pathology, the detailed molecular composition and regulatory pathways underlying exosome biogenesis and cargo selection remain unclear. Further research is needed to determine whether liraglutide directly influences the inclusion of specific bioactive molecules, such as microRNAs or proteins, within RPE-derived exosomes.

In summary, our study demonstrates that liraglutide mitigates FFA-induced oxidative stress, LD accumulation, EMT, and dysregulated exosome secretion in ARPE-19 cells. These effects are largely mediated through AMPK activation, underscoring its pivotal role in maintaining lipid homeostasis and RPE function. Furthermore, these findings provide new insights into the role of GLP-1 receptor signaling in RPE health and AMD progression. By addressing key pathological mechanisms, liraglutide shows promise as a potential therapeutic agent for AMD. While direct evidence showing that liraglutide can cross the BRB is currently lacking, studies have demonstrated its ability to penetrate the blood–brain barrier and exert neuroprotective effects in the central nervous system [39]. Additionally, other GLP-1 RAs, such as Exendin-4, have been shown to protect the BRB and reduce retinal vascular leakage in diabetic animal models [16]. These observations suggest that liraglutide may influence the retina either directly or indirectly, and they highlight the importance of further in vivo investigations to evaluate its delivery efficiency and retinal bioavailability. Notably, a recent large-scale cohort study [40] demonstrated that GLP-1 RAs, including liraglutide, are associated with a significantly reduced risk of both non-exudative and exudative AMD in an aging population, further supporting their translational potential. However, it is important to acknowledge that these findings are based on in vitro experiments. Additional studies in animal models are needed to validate these results and explore the long-term effects of liraglutide on RPE function within the complex retinal environment. Such investigations are essential to assess the translational potential of GLP-1 receptor agonists for AMD management.

## 4. Materials and Methods

### 4.1. Materials

All reagents were sourced from Sigma-Aldrich (St. Louis, MO, USA), unless otherwise noted. HCS LipidTox and Nile Red stains were purchased from Thermo Fisher Scientific (Waltham, MA, USA). Liraglutide was generously provided by Novo Nordisk (Bagsvaerd, Denmark). Primary antibodies against AMPK, phosphorylated Thr^172^-AMPK (pThr^172^-AMPK), mTOR, and phosphorylated Ser^2448^-mTOR (pSer^2448^-mTOR) were obtained from Cell Signaling Technology (Danvers, MA, USA). Additional antibodies for phosphorylated Ser-Smad2/3 and Smad2/3 were also procured from Thermo Fisher Scientific (Waltham, MA USA). Antibodies targeting E-cadherin, vimentin, ZO-1, and p62 were acquired from GeneTex (Irvine, CA, USA), and the PLIN2 antibody was sourced from ABclonal (Woburn, MA, USA). All chemicals were dissolved in phosphate-buffered saline (PBS) and stored at −20 °C until use.

### 4.2. Cell Culture and Viability Assay

ARPE-19 cells, an immortalized human retinal pigment epithelial cell line, were obtained from the American Type Culture Collection (ATCC, Manassas, VA, USA). Cells were routinely cultured in Dulbecco’s Modified Eagle Medium: Nutrient Mixture F-12 (DMEM/F-12, 1:1 mixture, Gibco, Thermo Fisher Scientific, Waltham, MA, USA) supplemented with 10% fetal bovine serum (FBS, Gibco) and 1% penicillin-streptomycin (100 U/mL penicillin, 100 μg/mL streptomycin, Gibco). Cultures were maintained at 37 °C in a humidified incubator with 5% CO_2_. For experimental procedures, cells were seeded in DMEM/F-12 supplemented with L-alanyl-L-glutamine (GlutaMAX, Gibco), 1% FBS, and antibiotics, at a density of 6 × 10⁴ cells/well in 24-well plates and incubated for 24 h prior to treatment. This reduced-serum medium was used to decrease cell proliferation and promote a more native retinal pigment epithelial phenotype, including polarized morphology and functional traits resembling native RPE cells [41]. For cell viability assays, cells were treated with varying concentrations of free fatty acids (FFA, oleic acid: palmitic acid = 2:1) or liraglutide for 24 h. Viability was assessed using the MTT assay, with absorbance measured at 570 nm using a microplate reader.

### 4.3. LipidTox Staining and Quantification Using High-Content Analysis (HCA)

Lipid droplet (LD) accumulation was assessed using LipidTox Green Neutral Lipid Stain (Thermo Fisher Scientific, Waltham, MA, USA) according to the manufacturer’s protocol. ARPE-19 cells were seeded in 24-well plates, treated under experimental conditions, and gently washed three times with PBS to remove residual media. LipidTox Green dye was diluted in PBS and incubated with cells for 30 min at 37 °C in the dark. After staining, cells were washed with PBS to eliminate unbound dye. Nuclear staining was performed using Hoechst 33342 (Thermo Fisher Scientific, Waltham, MA, USA) for enhanced visualization. Fluorescence imaging was conducted using an ImageXpress Micro Confocal High-Content imaging system (Molecular Devices, Sunnyvale, CA, USA). A minimum of 100 cells per condition were analyzed, with each experiment performed in triplicate. LD size and number were quantified using the MetaXpress software version 6.7.0.211 (Molecular Devices, Sunnyvale, CA, USA), which enabled automated, high-precision measurements and the detailed statistical analysis of lipid accumulation across all experimental groups.

### 4.4. Immunocytochemistry Staining

ARPE-19 cells were fixed with 4% paraformaldehyde for 15 min at room temperature and washed three times with PBS. Cells were then permeabilized using 0.1% Triton X-100 in PBS for 10 min. To prevent non-specific binding, they were blocked with 5% bovine serum albumin in PBS for 1 h at room temperature. Primary antibodies against ZO-1, E-cadherin, and vimentin were diluted 1:100 in the blocking solution and incubated overnight at 4° C. After washing three times with PBS, cells were incubated with Alexa Fluor-conjugated secondary antibodies (Thermo Fisher Scientific, Waltham, MA, USA) for 1 h at room temperature in the dark. Nuclei were counterstained with Hoechst 33342 for 10 min, followed by additional washes with PBS to remove excess dye. Fluorescence images were captured using an Olympus CKX-41 fluorescence microscope equipped with a DP74 camera (Olympus, Tokyo, Japan). Consistent imaging parameters were applied across all samples to ensure accurate and reproducible visualization of the target proteins.

### 4.5. Western Blot Analysis

Total protein lysates were prepared from ARPE-19 cells using Gold Lysis Buffer supplemented with protease and phosphatase inhibitors (Sigma-Aldrich, St. Louis, MO, USA) to preserve protein integrity. Protein concentrations were measured using the BCA protein assay kit (Thermo Fisher Scientific) to ensure equal loading for electrophoresis. Equal amounts of protein (20–40 µg per sample) were resolved by SDS-PAGE and transferred onto PVDF membranes (Millipore, Burlington, MA, USA). Membranes were blocked with 5% bovine serum albumin in TBST (Tris-buffered saline containing 0.1% Tween-20) for 1 h at room temperature to minimize non-specific binding. Primary antibodies were diluted 1:1000 in BSA-TBST and incubated with the membranes overnight at 4 °C. After three washes with TBST, membranes were incubated with HRP-conjugated secondary antibodies (diluted 1:10,000 in BSA-TBST) for 1 h at room temperature in the dark. Excess secondary antibodies were removed with three additional washes in TBST. Protein bands were visualized using an enhanced chemiluminescence (ECL) detection system (Bio-Rad, Hercules, CA, USA). Chemiluminescence signals were captured with an AI600 Imaging System (GE Healthcare, Chicago, IL, USA) under standardized exposure conditions. Relative protein expression levels were quantified using ImageJ software version 1.52a (NIH, Bethesda, MD, USA), with β-actin serving as the internal control. All experiments were performed in triplicate to ensure reproducibility and statistical reliability.

### 4.6. mRNA Expression Analysis Using qPCR

Total RNA was extracted from ARPE-19 cells using TRIzol reagent (Invitrogen, Carlsbad, CA, USA) following the manufacturer’s instructions. The concentration and purity of RNA were assessed using a spectrophotometer to ensure high-quality samples for downstream applications. Reverse transcription was performed using 1 µg of total RNA with a High-Capacity cDNA Reverse Transcription Kit (Applied Biosystems, Foster City, CA, USA) according to the manufacturer’s protocol. Quantitative PCR (qPCR) was conducted using SYBR Green Master Mix (Applied Biosystems, Foster City, CA, USA) on a 7300 Real-Time PCR System (Applied Biosystems). Each cDNA sample was analyzed in triplicate under the following cycling conditions: an initial denaturation at 95 °C for 10 min, followed by 40 amplification cycles of 95 °C for 15 s and 60 °C for 1 min. A dissociation curve was included at the end of the reaction to confirm the specificity of the amplification. Relative gene expression levels were determined using the 2^−^ΔΔCT method, with GAPDH serving as the internal reference gene for normalization. Primer sequences for target and housekeeping genes are listed in Table 1. Data analysis was performed using the Sequence Detection System software (v2.4.1, Applied Biosystems, Foster City, CA, USA), ensuring accurate and reproducible quantification.

### 4.7. Acidic Vesicular Organelles Detection via Acridine Orange Staining

Acridine orange (AO) staining was performed to assess autophagy by detecting acidic vesicular organelles (AVOs) in ARPE-19 cells. Cells were incubated with AO (1 μg/mL) for 15 min at 37 °C in the dark, followed by gentle washing with PBS to remove excess dye. Fluorescence signals were visualized using a fluorescence microscope, and images were captured under standardized settings to ensure consistency across samples. To quantify AVOs, HCA was conducted using an automated imaging system (Molecular Devices, Sunnyvale, CA, USA). A minimum of 100 cells per condition were analyzed, with each experiment performed in triplicate.

### 4.8. Quantification of Intracellular ROS Levels

Intracellular ROS levels were evaluated using the fluorescent dye DCFH-DA. Cells were incubated with 10 μM DCFH-DA for 30 min at 37 °C in the dark to allow dye uptake and oxidation. After incubation, cells were gently washed with PBS to remove excess dye. Stained cells were imaged using an Olympus CKX-41 fluorescence microscope equipped with a DP74 camera (Olympus, Tokyo, Japan) to capture fluorescence signals under consistent imaging conditions. Fluorescence intensity was quantified using the iD5 microplate reader (Molecular Devices, Sunnyvale, CA, USA) at excitation/emission wavelengths of 485/528 nm. For normalization, ROS levels in treated samples were quantified as a percentage relative to the untreated control. The fluorescence intensity of the untreated control samples was used to establish baseline ROS fluorescence, and ROS levels in treated samples were calculated using the following formula:$$Relative\;ROS\;Level\;(\%)=\frac{Fluorescence\;Intensity\;of\;Treated\;Sample}{Fluorescence\;Intensity\;of\;Control\;Sample}\times 100$$

Each treatment condition was assessed in triplicate to ensure reproducibility. This normalization approach facilitated the direct comparison of ROS levels across different treatments, with results expressed as a percentage relative to the control ROS level.

### 4.9. Assessment of Lipid Accumulation Using Nile Red Staining

Nile Red dye (1 μg/mL) was used to stain intracellular lipids for the assessment of lipid accumulation. Cells were incubated with the dye for 15 min at 37 °C in the dark. After incubation, fluorescence images were captured using an Olympus CKX-41 fluorescence microscope equipped with a DP74 camera (Olympus, Tokyo, Japan).

### 4.10. Extracellular Vesicle (EV) Isolation and Nanoparticle Tracking Analysis

EVs were isolated from conditioned media using qEV original 35 nm size exclusion columns (Izon Science, Christchurch, New Zealand). Before use, the columns were equilibrated to room temperature for 30 min. Conditioned media were pre-cleared by centrifugation at 300× *g* for 10 min to remove cells, followed by 10,000× *g* for 30 min to eliminate debris. The pellet was resuspended in 200 μL of PBS and loaded onto the qEV column. Fractions were collected using an automatic fraction collector-V2 (AFC-V2, Izon Science, Christchurch, New Zealand). The void volume (F0, 1 mL) was discarded, and fractions F1 through F7 (250 μL each) were collected per the manufacturer’s instructions. Fractions containing EVs were pooled and analyzed for size and concentration using a NanoSight NS300 system (Malvern Panalytical, Malvern, UK) equipped with a 488 nm laser. For the nanoparticle tracking analysis (NTA), samples were diluted 1:500 to 1:1000 in 0.22 µm-filtered PBS. The absence of background was confirmed using 0.2 µm-filtered PBS as a negative control. Camera level and detection threshold were set at 13 and 5, respectively. For each sample, four videos of 60 s each were recorded and analyzed using NTA 3.0 software (Malvern Panalytical, Malvern, UK). Particle size distribution and concentration data were obtained, and all measurements were conducted in triplicate to ensure accuracy and reproducibility.

### 4.11. Statistical Analysis

All experiments were conducted in triplicate, and data are presented as mean ± standard deviation (SD). Statistical significance was evaluated using one-way analysis of variance (ANOVA) followed by Tukey’s post hoc test for multiple comparisons. A *p*-value <0.05 was considered statistically significant. Data analysis was performed using SPSS Statistics 25 (SPSS, Inc., Chicago, IL, USA).

## Figures and Tables

**Figure 1 ijms-26-03704-f001:**
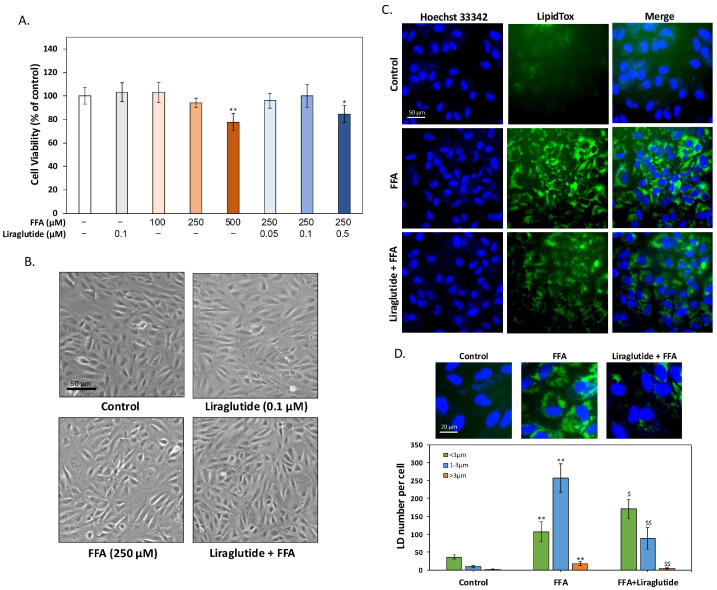
Liraglutide mitigates LD accumulation in ARPE-19 cells induced by FFA. (**A**) MTT assay results showing the effects of different concentrations of FFA and liraglutide on cell viability in ARPE-19 cells after 24 h. While 100 μM and 250 μM FFA did not affect cell viability, 500 μM significantly reduced viability. Liraglutide alone at 0.1 μM had no effect, but 0.5 μM liraglutide significantly reduced viability when combined with 250 μM FFA. Based on these results, 250 μM FFA and 0.1 μM liraglutide were used for subsequent experiments. (**B**) Bright-field images showed no visible changes in cell morphology or viability after treatment with 250 μM FFA, 0.1 μM liraglutide, or their combination. (**C**) LipidTox staining revealed significant intracellular LD accumulation in ARPE-19 cells treated with 250 μM FFA for 24 h. Co-treatment with 0.1 μM liraglutide effectively reduced LD accumulation. (**D**) High-content analysis (HCA) quantified LD size and distribution. FFA treatment significantly increased the number of LDs larger than 1 μm in diameter. Co-treatment with liraglutide reduced the total number of LDs and those ≥1 μm, while increasing the number of smaller LDs (<1 μm). These results suggest that liraglutide promotes LD degradation and mitigates FFA-induced lipid overload. All experiments were independently repeated at least three times. For HCA analysis, a minimum of 100 cells per condition were analyzed to ensure statistical robustness. All data are presented as mean ± SD. An asterisk (*) indicates a significant difference compared with the control group (* *p* < 0.05 and ** *p* < 0.01), and a dollar sign ($) indicates a significant difference compared with the FFA-treatment group ($ *p* < 0.05 and $$ *p* < 0.01).

**Figure 2 ijms-26-03704-f002:**
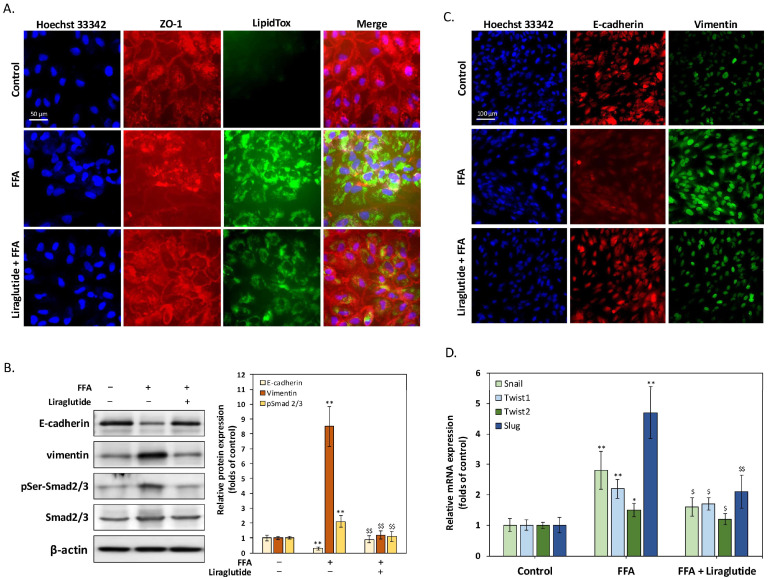
Liraglutide mitigates tight junction disruption and EMT in ARPE-19 cells induced by FFA. (**A**) Immunocytochemistry staining of ZO-1, a tight junction marker, in ARPE-19 cells treated with FFA (250 μM) and/or liraglutide (0.1 μM) for 24 h. FFA treatment significantly reduced ZO-1 expression at the cell membrane, particularly in regions with increased LD accumulation, as shown by LipidTox staining. Co-treatment with liraglutide restored ZO-1 expression to control levels. Scale bar = 100 μm. (**B**) Western blot analysis of epithelial marker E-cadherin, mesenchymal marker vimentin, and EMT-associated phosphorylated Smad2/3. FFA treatment reduced E-cadherin expression and increased vimentin and pSer-Smad2/3 levels. Liraglutide co-treatment reversed these changes, restoring E-cadherin and reducing vimentin and pSer-Smad2/3 expression. β-actin was used as a loading control. The bar graph shows relative protein expression normalized to β-actin. (**C**) Immunocytochemistry staining of E-cadherin and vimentin further supports the Western blot results. FFA treatment reduced E-cadherin and increased vimentin expression, both of which were reversed with liraglutide co-treatment. Scale bar = 100 μm. (**D**) Quantitative PCR analysis of EMT-related transcription factors, including Snail, Twist1, Twist2, and Slug. FFA significantly upregulated these transcription factors, indicating EMT induction, while liraglutide co-treatment suppressed their expression. All data were collected from at least three independent experiments and are presented as mean ± SD. An asterisk (*) indicates a significant difference compared with the control group (* *p* < 0.05 and ** *p* < 0.01), and a dollar sign ($) indicates a significant difference compared with the FFA-treatment group ($ *p* < 0.05 and $$ *p* < 0.01).

**Figure 3 ijms-26-03704-f003:**
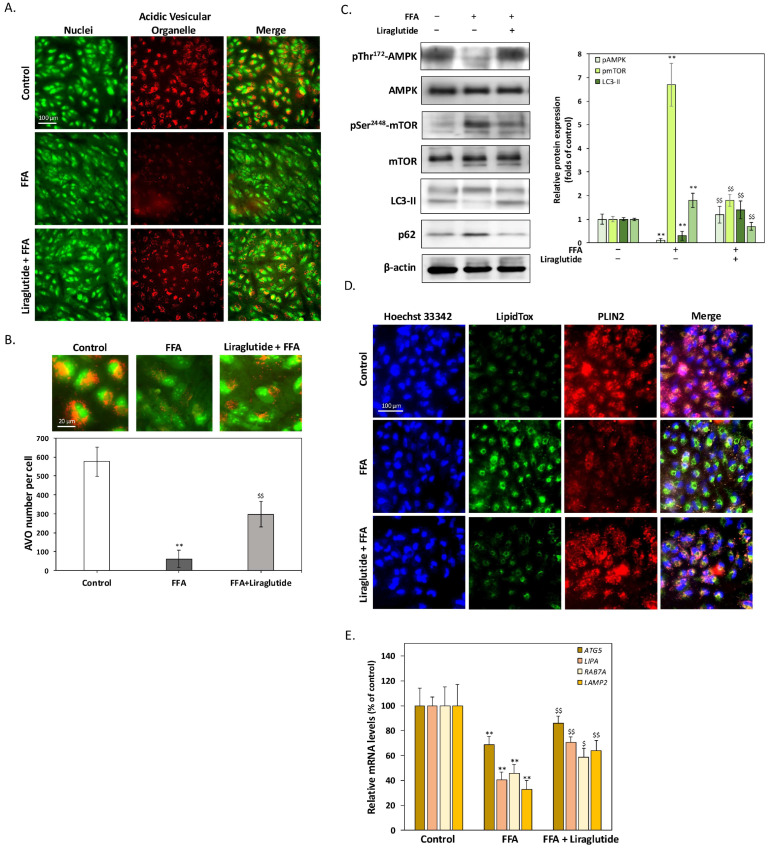
Liraglutide counteracts FFA-induced autophagy suppression and enhances lipophagy in ARPE-19 cells. (**A**) AO staining of ARPE-19 cells treated with FFA (250 μM) and/or liraglutide (0.1 μM) for 24 h. FFA significantly reduced the expression of AVOs, indicating suppressed autophagy, while liraglutide co-treatment restored AVO expression. Scale bar = 50 μm. (**B**) HCA quantification of AVOs. FFA reduced the number of AVOs to ~10% of control levels, while liraglutide co-treatment recovered AVOs to ~50% of control levels. (**C**) Western blot analysis of pThr172-AMPK, LC3-II, p62, and pSer^2448^-mTOR in ARPE-19 cells. FFA significantly decreased pThr^172^-AMPK and LC3-II expression and increased p62 levels, consistent with autophagy suppression. Additionally, FFA markedly elevated p-mTOR levels, indicating mTOR activation and the further inhibition of autophagy. Liraglutide co-treatment reversed these effects, restoring pThr^172^-AMPK and LC3-II levels while reducing p62 accumulation, and suppressing mTOR phosphorylation, suggesting the activation of AMPK-mediated autophagy through inhibition of the mTOR pathway. (**D**) Immunocytochemistry staining of PLIN2 in ARPE-19 cells. FFA markedly reduced PLIN2 expression, while liraglutide restored it. Co-localization of PLIN2 with LDs suggests that liraglutide promotes lipophagy and enhances LD degradation. Scale bar = 20 μm. (**E**) Quantitative PCR analysis of lipophagy and lysosomal degradation-related genes, including autophagy related 5 (ATG5), lysosomal acid lipase (LIPA), ras-related protein 7A (RAB7A), and lysosomal-associated membrane protein 2 (LAMP2). FFA significantly downregulated these genes, indicating suppressed LD degradation pathways, whereas liraglutide restored their expression levels. All data were collected from at least three independent experiments and are presented as mean ± SD. For the HCA quantification of AVOs, a minimum of 100 cells per condition were analyzed to ensure statistical robustness. An asterisk (*) indicates a significant difference compared with the control group (** *p* < 0.01), and a dollar sign ($) indicates a significant difference compared with the FFA-treatment group ($ *p* < 0.05 and $$ *p* < 0.01).

**Figure 4 ijms-26-03704-f004:**
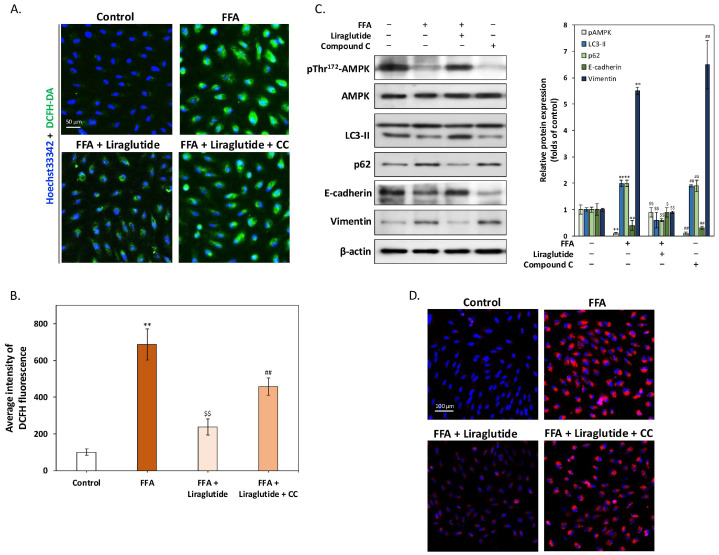
AMPK activation mediates liraglutide’s protective role in ARPE-19 cells under FFA-induced stress. (**A**) DCFH-DA staining of ARPE-19 cells treated with FFA (250 μM), liraglutide (0.1 μM), and/or compound C (CC, 10 μM) for 24 h. FFA significantly increased intracellular ROS, as indicated by enhanced DCFH fluorescence. Liraglutide reduced ROS accumulation, while CC co-treatment abrogated this effect. Scale bar = 50 μm. (**B**) The quantification of ROS levels using a fluorescence microplate reader. Intracellular ROS levels were quantified by measuring the fluorescence intensity of the DCFH-DA dye at excitation/emission wavelengths of 485/528 nm. FFA increased ROS levels approximately seven-fold compared to controls, whereas liraglutide reduced ROS by two-thirds. This reduction was reversed by CC co-treatment. (**C**) Western blot analysis of autophagy markers (pThr^172^-AMPK, LC3-II, p62) and EMT markers (E-cadherin, vimentin). FFA suppressed phosphorylated AMPK and LC3-II levels while increasing p62, indicating reduced autophagy. It also decreased E-cadherin and increased vimentin, confirming EMT induction. Liraglutide reversed these effects, but CC co-treatment reduced liraglutide’s protective impact. (**D**) Nile red staining of intracellular lipid. FFA increased lipid accumulation, while liraglutide reduced intracellular lipid levels. CC co-treatment reversed liraglutide’s effect on lipid reduction. Scale bar = 100 μm. All data were collected from at least three independent experiments and are presented as mean ± SD. An asterisk (*) indicates a significant difference compared to the control group (** *p* < 0.01), a dollar sign ($) indicates a significant difference compared to the FFA-treatment group ($ *p* < 0.05 and $$ *p* < 0.01), and a pound sign (#) indicates a significant difference compared to the FFA and liraglutide co-treatment group (## *p* < 0.01).

**Figure 5 ijms-26-03704-f005:**
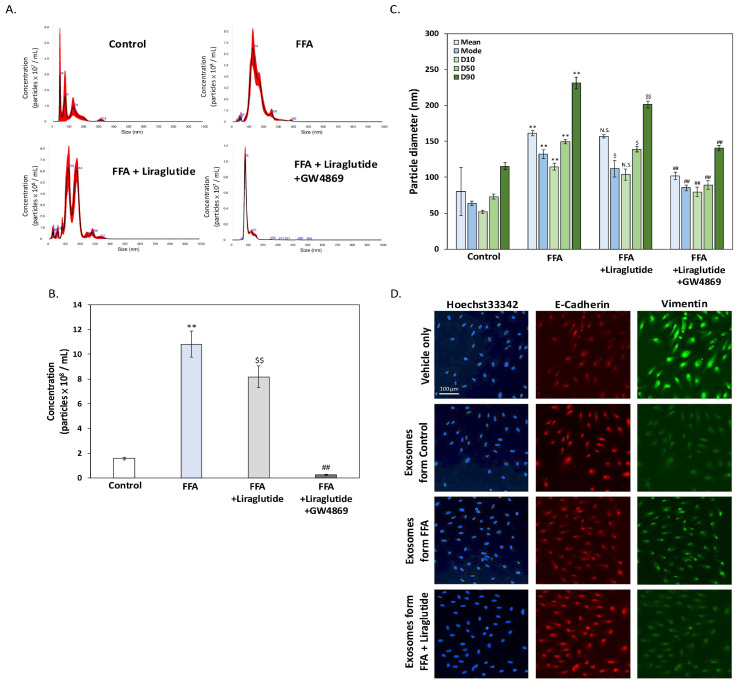
Liraglutide regulates exosome characteristics in ARPE-19 cells. (**A**) The NTA results of EVs secreted by ARPE-19 cells under the following treatment conditions are recorded: control, FFA (250 μM), FFA + liraglutide (0.1 μM), and FFA + liraglutide + GW4869 (10 μM). FFA significantly increased EV secretion, while liraglutide reduced this effect. GW4869, an exosome biogenesis/release inhibitor, drastically decreased EV secretion. (**B**) Quantification of EV concentration from NTA data. FFA increased EV concentration approximately 10-fold compared to the control. Liraglutide reduced EV levels to approximately eight times the control, while GW4869 significantly suppressed EV secretion. (**C**) Analysis of EV particle size distribution from NTA data revealed that FFA significantly increased the average diameter of EVs. Liraglutide did not alter the mean diameter but significantly reduced the median diameter (D50), the 90th percentile (D90), and the mode diameter of EVs. (**D**) EVs isolated from conditioned media were standardized to a concentration of 5 × 10⁷ particles/mL and were co-cultured with FFA-treated ARPE-19 cells for 24 h. Immunocytochemistry analysis revealed that EVs derived from the control and FFA + liraglutide groups enhanced E-cadherin expression and suppressed vimentin expression, indicating anti-EMT effects. EVs from the FFA group also reduced EMT but to a lesser extent compared to the control and FFA + liraglutide groups. Scale bar = 100 μm. All data were collected from at least three independent experiments and are presented as mean ± SD. An asterisk (*) indicates a significant difference compared to the control group (** *p* < 0.01), a dollar sign ($) indicates a significant difference compared to the FFA-treatment group ($ *p* < 0.05 and $$ *p* < 0.01), and a pound sign (#) indicates a significant difference compared to the FFA and liraglutide co-treatment group (## *p* < 0.01).

**Table 1 ijms-26-03704-t001:** Primer sequence of different genes for qPCR analysis.

Genes	Forward (5′-3′)	Reverse (5′-3′)
*S* *NAI1*	CCTCCCTGTCAGATGAGGAC	CCAGGCTGAGGTATTCCTTG
*T* *WIST* *1*	GCCAGGTACATCGACTTCCTCT	TCCATCCTCCAGACCGAGAAGG
*T* *WIST* *2*	GAGTCCGCAGTCTTACGAGG	GCTCTGCAGCTCCTCGTCT
*ATG5*	GGAATGGTGTTCGTCCTTCA	CAGGGAAGGAACAGCTTTGA
*LIPA*	GTGGGTCATTCTCAAGGCACCA	CCATAGGGCTAGTACAGAAGGC
*RAB7A*	CCTCGAAAACAGACAAGTGGC	ATTCCGTGCAATCGTCTGGA
*LAMP2*	TATGTGCAACAAAGAGCAGA	CAGCATGATGGTGCTTGAG
*GAPDH*	GATTCCACCCATGGCAAATTC	CTGGAAGATGGTGATGGGATT

## Data Availability

The data that support the findings of this study are available from the corresponding author, Lin C.-L. and Hsu M.-Y., upon reasonable request due to privacy or ethical restrictions.

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
