# Peer review of "Liraglutide Attenuates FFA-Induced Retinal Pigment Epithelium Dysfunction via AMPK Activation and Lipid Homeostasis Regulation in ARPE-19 Cells"

_ijms, 2025, doi:10.3390/ijms26083704_

Round 1

Reviewer 1 Report

Comments and Suggestions for Authors

The authors utilized APRE-19 cells to investigate the protective effects of Glucagon-like peptide-1 receptor agonists (GLP-1 RAs) against free fatty acid-induced cytotoxicity. The manuscript presents several interesting findings, including the beneficial effects of extracellular vesicles produced by liraglutide-treated cells. Western blot analysis revealed changes in AMPK activation and autophagy markers, with the results further supported by quantitative imaging data. Overall, the study is well-designed and will likely be of great interest to those researching GLP-1 agonists. There are a few suggestions to enhance the manuscript:

  1. The manuscript currently presents the results section before the methods section. Please provide some experimental details related to Figure 3, and ensure that the full names of abbreviations are given when they first appear in the text.
  2. It would be valuable to include a discussion on the controversies surrounding the potential adverse effects of GLP-1 receptor agonists in diabetic retinopathy (PMID 31227789). This could further strengthen the importance of understanding the underlying mechanisms.
  3. Most previous research has used ARPE-19 cells to demonstrate the expression of the GLP-1 receptor. Could the authors provide in vivo evidence, such as data from existing single-cell RNA sequencing databases, to show that retinal pigment epithelial (RPE) cells express this receptor?
  4. Would the authors be able to provide data on changes in mTOR activity following treatments? Since mTOR is downstream of AMPK, alterations in the mTOR pathway may help explain the metabolic dysfunctions observed in the RPE.

Author Response

Reviewer 1

The authors utilized APRE-19 cells to investigate the protective effects of Glucagon-like peptide-1 receptor agonists (GLP-1 RAs) against free fatty acid-induced cytotoxicity. The manuscript presents several interesting findings, including the beneficial effects of extracellular vesicles produced by liraglutide-treated cells. Western blot analysis revealed changes in AMPK activation and autophagy markers, with the results further supported by quantitative imaging data. Overall, the study is well-designed and will likely be of great interest to those researching GLP-1 agonists. There are a few suggestions to enhance the manuscript:

The manuscript currently presents the results section before the methods section. Please provide some experimental details related to Figure 3, and ensure that the full names of abbreviations are given when they first appear in the text.

Ans: We sincerely thank the reviewer for the thoughtful and constructive comments. In response, we have revised the manuscript to clarify the experimental details related to Figure 3. In addition, we have provided the full names of all abbreviations at their first appearance in the text. Specifically, "AO" (acridine orange), "AVOs" (acidic vesicular organelles), and "HCA" (high-content analysis) are now fully defined, with "HCA" introduced earlier in the Results section under Figure 1.

It would be valuable to include a discussion on the controversies surrounding the potential adverse effects of GLP-1 receptor agonists in diabetic retinopathy (PMID 31227789). This could further strengthen the importance of understanding the underlying mechanisms.

Ans: We appreciate the reviewer’s insightful suggestion regarding the potential adverse effects of GLP-1 receptor agonists on diabetic retinopathy. In response, we have incorporated a discussion of the relevant findings from PMID: 31227789 (Ref#35) at the end of the second paragraph in the Discussion section. This addition highlights the reported risk of retinopathy complications associated with semaglutide and underscores that, despite the observed therapeutic effects of liraglutide in our study, GLP-1 receptor agonists may still pose potential risks to retinal health. Therefore, further in-depth investigation is warranted to fully elucidate their safety profile in the context of AMD.

Most previous research has used ARPE-19 cells to demonstrate the expression of the GLP-1 receptor. Could the authors provide in vivo evidence, such as data from existing single-cell RNA sequencing databases, to show that retinal pigment epithelial (RPE) cells express this receptor?

Ans: We thank the reviewer for raising this important point. In response, we have cited the study by Fan et al. (Exp Eye Res, 2014; Ref#16), which provides in vivo evidence that GLP-1 receptor (GLP-1R) is expressed in the retina, including the retinal pigment epithelium (RPE). Immunohistochemical analyses in diabetic Goto-Kakizaki rats clearly demonstrated GLP-1R expression in multiple retinal layers, supporting our use of RPE cells as a relevant model for investigating GLP-1R signaling.

Would the authors be able to provide data on changes in mTOR activity following treatments? Since mTOR is downstream of AMPK, alterations in the mTOR pathway may help explain the metabolic dysfunctions observed in the RPE.

Ans: We thank the reviewer for this insightful comment. In response, we have added new Western blot analyses of mTOR expression and phosphorylation levels in Figure 3. These results demonstrate that liraglutide-induced AMPK activation leads to a significant suppression of mTOR activity in FFA-treated ARPE-19 cells. This supports the notion that the AMPK/mTOR axis is involved in the regulation of lipophagy and may underlie the observed improvements in lipid clearance and cellular homeostasis in RPE cells.

Reviewer 2 Report

Comments and Suggestions for Authors

The study explores the potential of liraglutide, a GLP-1 receptor agonist, in mitigating free fatty acid (FFA)-induced damage in ARPE-19 cells, which serve as a model for retinal pigment epithelium (RPE) dysfunction relevant to age-related macular degeneration (AMD). The research is well-structured, presenting strong experimental support for its conclusions. However, some areas require further clarification and additional discussion.

Minor revision

  1. Clarification of the Therapeutic Relevance
    • The study primarily focuses on in vitro findings. While these are compelling, it would be beneficial to discuss whether liraglutide can cross the blood-retina barrier and how it could be delivered effectively in in vivo settings.
    • A brief mention of whether liraglutide or other GLP-1 receptor agonists have been tested in animal models of AMD would enhance translational relevance.
  2. Elaboration on Exosome Findings
    • The study reports that liraglutide alters exosome secretion and size distribution. However, the functional implications of these changes remain unclear.
    • Did the authors analyze the molecular cargo of exosomes?
    • Could liraglutide-modulated exosomes be mediating additional protective effects beyond reducing EMT markers?
  3. Consideration of Potential Off-Target Effects
    • The study attributes the effects of liraglutide to AMPK activation, but liraglutide may also influence other metabolic pathways in RPE cells.
    • Did the authors evaluate whether liraglutide affects other signaling cascades (e.g., mTOR, SIRT1), which could also regulate autophagy and lipid metabolism?
  4. Figure Readability and Statistical Transparency
    • Some figures, particularly those related to Western blots and immunofluorescence staining, could be optimized for clarity. Enlarging labels and ensuring uniform brightness/contrast adjustments would improve readability.
    • While p-values are reported, effect sizes or confidence intervals would strengthen the statistical rigor of the conclusions.
  5. Discussion of Alternative Pathways for EMT Regulation
    • While the study highlights AMPK’s role in EMT suppression, other pathways such as TGF-β/Smad signaling are known to regulate EMT in RPE cells.

Could liraglutide be exerting indirect effects on these pathways?

Author Response

Reviewer 2

The study explores the potential of liraglutide, a GLP-1 receptor agonist, in mitigating free fatty acid (FFA)-induced damage in ARPE-19 cells, which serve as a model for retinal pigment epithelium (RPE) dysfunction relevant to age-related macular degeneration (AMD). The research is well-structured, presenting strong experimental support for its conclusions. However, some areas require further clarification and additional discussion.

Minor revision

Clarification of the Therapeutic Relevance

The study primarily focuses on in vitro findings. While these are compelling, it would be beneficial to discuss whether liraglutide can cross the blood-retina barrier and how it could be delivered effectively in in vivo settings.

Ans: We appreciate the reviewer’s valuable comment regarding the potential in vivo applicability of liraglutide. While direct evidence demonstrating the ability of liraglutide to cross the blood-retina barrier (BRB) is currently lacking, previous studies have shown that liraglutide can cross the blood-brain barrier (BBB) and exert neuroprotective effects within the central nervous system (Front Neurol

2024;15:1462240, Ref#40). Moreover, other GLP-1 receptor agonists, such as Exendin-4, have been reported to protect the BRB and reduce retinal vascular leakage in diabetic animal models (Exp Eye Res 2014 127;104-16, Ref#16). These findings suggest that liraglutide may similarly influence the retina, either by crossing the BRB or by modulating systemic pathways. In accordance with the reviewer’s suggestion, we have incorporated this discussion into the revised manuscript to strengthen the interpretation of our findings and address the translational potential of liraglutide.

A brief mention of whether liraglutide or other GLP-1 receptor agonists have been tested in animal models of AMD would enhance translational relevance.

Ans: We thank the reviewer for this helpful suggestion. In response, we have added a brief discussion referencing a recent large-scale retrospective cohort study (Allan et al., 2025, Ref#41) that evaluated the use of GLP-1 receptor agonists, including liraglutide, in an at-risk aging population. This study demonstrated that GLP-1 RAs were associated with a significantly reduced hazard of both non-exudative and exudative AMD. In addition, prior animal studies have shown that GLP-1 RAs exert neuroprotective and anti-inflammatory effects in the retina. These findings support the potential translational relevance of our in vitro results, and we have included this reference in the revised manuscript.

Elaboration on Exosome Findings

The study reports that liraglutide alters exosome secretion and size distribution. However, the functional implications of these changes remain unclear.

Did the authors analyze the molecular cargo of exosomes?

Ans: We thank the reviewer for this important question. While we did not analyze the molecular cargo of exosomes in the current study, we agree that this aspect is critical for understanding their functional roles. In our recent clinical research (Wu et al., 2024, Ref#39), we observed that exosome size distributions in aqueous humor samples vary significantly across different retinal diseases and treatment statuses. Specifically, larger exosomes (90–120 nm) were more prevalent in inflammatory conditions such as pars planitis, while smaller exosomes or exomeres (<50 nm) were dominant in ischemic conditions like central retinal vein occlusion. These findings suggest that exosome size may reflect disease-specific pathophysiological states and could potentially influence their cargo composition and biological functions. We have added this discussion to the revised manuscript to highlight the need for future studies analyzing exosomal content in relation to their size and disease context.

Could liraglutide-modulated exosomes be mediating additional protective effects beyond reducing EMT markers?

Ans: We sincerely thank the reviewer for this insightful comment. We agree that liraglutide-modulated exosomes may mediate additional protective effects beyond reducing EMT markers, such as influencing inflammation, oxidative stress, or angiogenesis. Although our current study focused primarily on EMT regulation, we recognize the potential for broader exosome-mediated mechanisms. Further investigation is warranted to characterize the exosomal cargo and explore these additional effects. We truly appreciate the reviewer’s suggestion

Consideration of Potential Off-Target Effects

The study attributes the effects of liraglutide to AMPK activation, but liraglutide may also influence other metabolic pathways in RPE cells.

Did the authors evaluate whether liraglutide affects other signaling cascades (e.g., mTOR, SIRT1), which could also regulate autophagy and lipid metabolism?

Ans: We thank the reviewer for this valuable comment. We agree that liraglutide may influence multiple signaling pathways beyond AMPK, including those such as mTOR and SIRT1, which are also critically involved in regulating autophagy and lipid metabolism. In this revised version of the manuscript, we have added new Western blot data to evaluate the impact of liraglutide on mTOR signaling (see Figure 3). Our results demonstrate that liraglutide significantly suppresses mTOR phosphorylation in FFA-treated ARPE-19 cells, supporting the hypothesis that liraglutide’s effects are mediated, at least in part, through the AMPK/mTOR axis. While SIRT1 was not assessed in the current study, we appreciate the reviewer’s suggestion and will consider exploring this and other pathways in future investigations to further delineate the broader metabolic network modulated by liraglutide in RPE cells.

Figure Readability and Statistical Transparency

Some figures, particularly those related to Western blots and immunofluorescence staining, could be optimized for clarity. Enlarging labels and ensuring uniform brightness/contrast adjustments would improve readability.

While p-values are reported, effect sizes or confidence intervals would strengthen the statistical rigor of the conclusions.

Ans: We sincerely thank the reviewer for the helpful suggestions regarding figure clarity and statistical reporting. In response, we have revised the figures related to Western blotting and immunofluorescence staining by enlarging labels and adjusting the brightness and contrast to ensure consistent image presentation and improved readability. Regarding statistical rigor, while our original data included p-values to indicate significance, we acknowledge the importance of providing more comprehensive statistical metrics. Therefore, we have now included effect sizes where applicable in the figure legends. These revisions aim to enhance both the visual clarity and scientific transparency of our study.

Discussion of Alternative Pathways for EMT Regulation

While the study highlights AMPK’s role in EMT suppression, other pathways such as TGF-β/Smad signaling are known to regulate EMT in RPE cells.

Could liraglutide be exerting indirect effects on these pathways?

Ans: We thank the reviewer for this thoughtful and important comment. Indeed, TGF-β/Smad signaling is a well-established pathway involved in EMT regulation in RPE cells, and we agree that it may interact with or be influenced by other signaling cascades such as AMPK. Although our study focused primarily on AMPK-mediated mechanisms, we acknowledge the possibility that liraglutide may exert indirect effects on the TGF-β/Smad pathway. Previous reports have suggested that AMPK activation can negatively regulate TGF-β signaling, potentially through Smad2/3 inhibition, thereby contributing to EMT suppression (Mol Pharmacol. 2015; 88:1062–1071). While we did not specifically evaluate TGF-β/Smad components in this study, we appreciate the reviewer’s suggestion and will explore this alternative regulatory axis in future investigations to further delineate the broader signaling landscape modulated by liraglutide.

Round 2

Reviewer 1 Report

Comments and Suggestions for Authors

all previous comments were well addressed

Author Response

In response to academic editor's comments:

  1. We have revised the legend of Figure 3 to include a reference to mTOR and pSer2448-mTOR in Panel C, in alignment with the newly added Western blot images in the revised version of our manuscript.
  2. Regarding the legends of Figures 4 and 5, we did not add references to the number of cells evaluated in the immunostaining. For Fig. 4B, the ROS quantification was based on measuring total fluorescence intensity per well using a microplate reader, which does not involve cell-level evaluation. For Fig. 5D, as no quantitative analysis was conducted on these immunostaining results.